# Transcytosable Peptide-Paclitaxel Prodrug Nanoparticle for Targeted Treatment of Triple-Negative Breast Cancer

**DOI:** 10.3390/ijms24054646

**Published:** 2023-02-28

**Authors:** Longkun Wang, Chunqian Zhao, Lu Lu, Honglei Jiang, Fengshan Wang, Xinke Zhang

**Affiliations:** 1Key Laboratory of Chemical Biology (Ministry of Education), NMPA Key Laboratory for Quality Research and Evaluation of Carbohydrate-Based Medicine, Institute of Biochemical and Biotechnological Drug, School of Pharmaceutical Sciences, Cheeloo College of Medicine, Shandong University, Jinan 250012, China; 2Key Laboratory of Chemical Biology (Ministry of Education), Department of Pharmacology, School of Pharmaceutical Sciences, Cheeloo College of Medicine, Shandong University, Jinan 250012, China

**Keywords:** paclitaxel, peptide-drug conjugate, TAT peptide, A7R peptide, targeted delivery, anticancer therapy, transcytosis

## Abstract

Triple-negative breast cancer (TNBC) is an extremely aggressive subtype associated with a poor prognosis. At present, the treatment for TNBC mainly relies on surgery and traditional chemotherapy. As a key component in the standard treatment of TNBC, paclitaxel (PTX) effectively inhibits the growth and proliferation of tumor cells. However, the application of PTX in clinical treatment is limited due to its inherent hydrophobicity, weak penetrability, nonspecific accumulation, and side effects. To counter these problems, we constructed a novel PTX conjugate based on the peptide-drug conjugates (PDCs) strategy. In this PTX conjugate, a novel fused peptide TAR consisting of a tumor-targeting peptide, A7R, and a cell-penetrating peptide, TAT, is used to modify PTX. After modification, this conjugate is named PTX-SM-TAR, which is expected to improve the specificity and penetrability of PTX at the tumor site. Depending on hydrophilic TAR peptide and hydrophobic PTX, PTX-SM-TAR can self-assemble into nanoparticles and improve the water solubility of PTX. In terms of linkage, the acid- and esterase-sensitive ester bond was used as the linking bond, with which PTX-SM-TAR NPs could remain stable in the physiological environment, whereas PTX-SM-TAR NPs could be broken and PTX be released at the tumor site. A cell uptake assay showed that PTX-SM-TAR NPs were receptor-targeting and could mediate endocytosis by binding to NRP-1. The vascular barrier, transcellular migration, and tumor spheroids experiments showed that PTX-SM-TAR NPs exhibit great transvascular transport and tumor penetration ability. In vivo experiments, PTX-SM-TAR NPs showed higher antitumor effects than PTX. As a result, PTX-SM-TAR NPs may overcome the shortcomings of PTX and present a new transcytosable and targeted delivery system for PTX in TNBC treatment.

## 1. Introduction

Triple-negative breast cancer (TNBC) is a subtype of breast cancer that lacks the expression of estrogen and progesterone receptor (ER/PR), as well as human epidermal growth factor receptor (HER-2), accounting for about 20% of the total diagnosed breast cancers in the world [1,2]. TNBC is mainly prevalent in young women and has the characteristics of high invasion, high heterogeneity, and poor prognosis. The high heterogeneity and lack of specific receptors of TNBC limit the choice of clinical treatments. Nowadays, surgery combined with chemotherapy drugs is still the standard treatment for TNBC [3,4,5]. With the in-depth understanding of TNBC molecular subtyping, an increasing number of targeted drugs have entered research and development [6,7,8,9]. Up to now, four targeted drugs have been approved by the FDA for the treatment of TNBC, including two PARP1 inhibitors (olaparib and talazoparib), one programmed cell death 1 ligand (PD-L1) inhibitor (atezolizumab), and one antibody-drug conjugate (ADC) (sacituzumab-govitecan) [10]. Nevertheless, these drugs cannot meet the demand of all TNBC patients, and new targeting strategies need to be continuously developed to improve the clinical cure rate of TNBC [11].

Paclitaxel (PTX) is a common antitumor agent that mainly acts on cytoplasmic microtubules. PTX binds covalently to the β-subunit of a tubulin protein to interfere with the dynamic balance of microtubules and inhibit microtubule depolymerization, which would arrest cell division in the G2/M phase and form multinucleated cells, ultimately leading to cell death [12,13]. On account of its potent tumor inhibitory effect, paclitaxel has been used for a long time in the clinical treatment of various cancers. As a key component in the standard treatment of TNBC, PTX is an indispensable and important drug in the clinical treatment of TNBC [14]. However, due to its inherent hydrophobic property, weak penetrability, and nonspecific accumulation, the clinical application of PTX is limited to a certain extent [15]. To address these issues, many PTX formulations have been studied. So far, three PTX formulations have been approved by the FDA, including Taxol^®^, Abraxane^®^, and Xyotax^®^ [16,17,18]. Nevertheless, they only provide an appropriate solubilization system rather than an ideal active targeted drug delivery system, with a slight attenuation in systemic toxicity caused by PTX. To obtain better clinical applications of PTX, a variety of strategies have been explored continuously [19,20,21].

Peptide-drug conjugates (PDCs) are a kind of new drug delivery strategy derived from antibody-drug conjugates (ADCs), which are formed by coupling flexible peptides with small molecule drugs through cleavage or non-cleavage linkers [22,23,24]. The multifunctional peptides endow the conjugates with some excellent properties, including water solubility, targeting, sensitization, and penetrability [25,26,27,28]. In contrast to other drug delivery systems, PDCs have the advantages of small molecular weight, easy synthesis, low immunogenicity, and a flexible structure [29]. Consequently, PDCs gradually play an important role in tumor therapy. Peptides commonly used in PDCs are divided into two categories, tumor-targeting peptides (TTPs) and cell-penetrating peptides (CPPs) [30,31]. CPPs are widely used in drug delivery systems and can cross the cell membrane through energy-dependent or non-energy-dependent ways to enhance the cellular uptake of drugs [32]. TAT (RKKRRQRRR), as one of the most common CPPs, has the feature of efficient and non-invasive transmembrane transport [33,34]. Nevertheless, due to the lack of tumor specificity, TAT-induced systemic distribution may lead to reduced drug effects and enhanced side effects. To solve the problem, coupling CPPs with TTPs may be an effective method [35,36,37,38].

Solid tumor tissues not only contain tumor cells but also diffuse a large number of newly formed blood vessels. Angiogenesis, the growth of new blood vessels from existing vascular systems, plays a crucial role in cell nutrition supply and metabolic waste excretion. Therefore, angiogenesis is an essential event in tumor progression and is considered an attractive target for cancer therapy [39,40]. Neuropilin 1 (NRP-1) is an important receptor in tumor angiogenesis that regulates this process by binding to ligand VEGF-A and activating downstream signaling pathways [41,42]. NRP-1 is highly expressed in tumor vascular endothelial cells and some tumor cells [43,44]. Additionally, it has been confirmed that NRP-1 expression in TNBC is up-regulated compared to other breast cancer subtypes, and NRP-1 can be used as a potential target for TNBC treatment [45,46,47]. Hence, NRP-1 mediated targeted therapy may be a potential strategy for precise TNBC therapy by simultaneously targeting tumor cells and vascular endothelial cells. An A7R (ATWLPPR) peptide selected by phage screening technology can target NRP-1 and is widely used in tumor-targeted therapy [48].

In the previous study, we designed a novel bifunctional fusion peptide, including A7R and TAT, named TAR peptide. TAR peptide can also target NRP-1 for cargo delivery and effectively penetrate the tumor barrier, which has been used for DNA delivery for the treatment of glioma [49]. In this study, TAR peptide was used to deliver PTX into breast tumors more efficiently. TAR peptide was coupled with PTX using an acid- and esterase-sensitive ester bond and formed a novel prodrug conjugate PTX-SM-TAR. The main difference between tumor cells and normal cells lies in their morphological structure and the mode of growth and metabolism. Tumor cells grow and metabolize more vigorously, thus creating a microenvironment rich in lactic acid and various proteases. Therefore, the conjugate PTX-SM-TAR can remain stable in the normal physiological environment, whereas the ester bond can be degraded quickly and PTX can be released from PTX-SM-TAR at the tumor site with low pH and high esterase activity.

In addition, the conjugate PTX-SM-TAR synthesized by hydrophilic TAR peptide and hydrophobic PTX is amphiphilicity and has the characteristics of self-assembly in an aqueous solution to form a core-shell nano-micelle. In antitumor therapy, nanomedicines mainly rely on the enhanced permeability and retention effect (EPR) to target the tumor tissue passively. However, the EPR effect is hindered by increased tumor interstitial fluid pressure, high viscous extracellular matrix, and tumor heterogeneity, which leads to the limitation of drug extravasation in blood vessels and insufficient penetration into the tumor entity to maximize the efficacy of nanomedicines [50]. Transcytosis is a broad transcellular transport process in which substances are transferred from one side of a cell to the other. Unlike passive diffusion, transcytosis is an active transport process that can deliver drugs through multiple pathways [51]. In recent years, transcytosable nanomedicine has become a new interest in promoting the penetration of nanomedicines in tumors [52,53]. The transcytosis design of these nanomedicines could depend on different forms, including receptor-mediated transcytosis, adsorption-mediated transcytosis, and fluid-mediated transcytosis [51]. Based on the receptor-binding and positive charge properties of TAR peptides, we hypothesized that in addition to the EPR effect, the PTX-SM-TAR nanosystem can also be rapidly internalized by endothelial cells or tumor cells through receptor and adsorption-mediated transcytosis, improving extravasation efficiency in tumor vessels and enhancing permeability in tumor tissues. In this study, we focus on developing a novel prodrug conjugate using the PDCs strategy and evaluating its efficacy in the treatment of TNBC. The prodrug conjugate PTX-SM-TAR was designed and synthesized by linking hydrophobic PTX to the hydrophilic TAR peptide using an ester bond. The PTX-SM-TAR possessed the following advantages: (1) With amphiphilic properties, it can self-assemble into nano-complexes, which improves the water solubility of PTX. (2) With an acid- and esterase-sensitive linker, the conjugate can quickly release PTX under certain circumstances. (3) TAR peptide gives the conjugate excellent tumor targeting, transvascular transport, and penetration properties, which could deliver more PTX to arrive at the TNBC tumor site and take active effects. In summary, the conjugate PTX-SM-TAR might be a potential targeted therapy for TNBC.

## 2. Results

### 2.1. The Targeting and Drug Delivery Properties of TAR Peptide

The binding affinity of the TAR peptide to NRP-1 was measured with an SRP assay. As shown in Appendix A, TAR showed a strong binding ability to NRP-1 with a K_D_ value of 3.397 × 10^−8^ M. TAR peptide labeled with fluorescein Cy5.5-MAL was used to analyze the tumor-targeting delivery ability of a TAR peptide in vivo. After an intravenous injection of Cy5.5-MAL-TAR and free Cy5.5-MAL via the tail vein, the entire body and ex vivo fluorescence images were recorded using an in vivo imaging system. In tumor tissues, strong fluorescence was detected in the Cy5.5-MAL-TAR group while only modest or no fluorescence was detected in the Cy5.5-MAL group (Figure 1a,b). After 8 h of injection, the mice were sacrificed, and the main tissues (heart, liver, spleen, lung, kidney, and tumor) were collected for in vitro fluorescence detection (Figure 1c). The fluorescence intensity of the two groups was equivalent in the spleen and kidney. The fluorescence of Cy5.5-MAL-TAR was significantly weakened in the heart, liver, and lung tissues, especially in lung tissues, which is 40% of free Cy5.5-MAL. However, Cy5.5-MAL-TAR exhibited stronger fluorescence at the tumor site, 1.64 times that of free Cy5.5-MAL (Figure 1d). The slice analysis of tumor tissues showed that Cy5.5-MAL-TAR was widely dispersed in tumor tissues covering all micro-vessels and tumor cells that over-expressing NRP-1 while free Cy5.5-MAL only diverged in a spot-like manner and had no correlation with the expression level of NRP-1 (Figure 1e,f).

### 2.2. Synthesis and Characterization of PTX-SM-TAR

The synthetic route of PTX-SM-TAR was shown in Figure 2a, which was carried out in three steps. With SA-MAL as the connecting arm, PTX-SA was firstly formed by esterification at the 2′-OH position of PTX, and then PTX-SA-MAL was synthesized by amide reaction. Finally, the equivalent amount of PTX-SA-MAL and the TAR peptide were coupled together through a thioether bond. These products were characterized by ^1^H-NMR and Q-TOF HRMS spectrum after purification (Appendix A). As shown in Figure 2b, the characteristic peak of the secondary amine of tryptophan in the TAR peptide sequence at 10.834 ppm and that of the secondary amine in the PTX structure at 9.239 ppm. The ratio of the two peak areas (1:1) was consistent with the structural characteristics of PTX-SM-TAR. In the Q-TOF HRMS spectrum, the peak signal of 1671.3668 ([M+2H]^2+^) and 1114.5837 ([M+3H]^3+^) were consistent with the calculated molecular weight (3340.8680), suggesting that PTX-SM-TAR has been successfully synthesized (Figure 2c). The purity of PTX-SM-TAR was determined to be 95% by analytical HPLC (Figure 2d).

### 2.3. CAC, Morphology, Size, and Zeta Potential of PTX-SM-TAR NPs

The PTX-SM-TAR conjugate is a typical amphiphilic molecule composed of a hydrophobic PTX and a hydrophilic TAR peptide. Based on the molecular properties, PTX-SM-TAR conjugate can self-assemble into nanoparticles in an aqueous environment. The CAC of PTX-SM-TAR NPs was determined by the conductivity method to be 3.95 × 10^−6^ mol/L (Figure 3a). TEM and DLS were performed to characterize the morphology, particle size, size distribution, and zeta potential of PTX-SM-TAR NPs. DLS results showed that the PTX-SM-TAR conjugate could form uniform nanocomposites (PDI = 0.21 ± 0.03) with a particle size around 94.70 ± 1.87 nm (Figure 3b) and zeta potential around 14.53 ± 0.31 mV (Figure 3c). TEM results displayed that PTX-SM-TAR conjugate can form spherical nanoparticles in an aqueous solution (Figure 3d).

### 2.4. In Vitro PTX Release from PTX-SM-TAR NPs

In order to evaluate the sensitivity of PTX-SM-TAR NPs to acid and esterase, the in vitro release of free PTX was investigated in a PBS buffer containing 1% Tween 80. Here, PBS at pH 7.4 mimics the normal physiological environment while PBS at pH 6.8 with esterase mimics the microenvironment around the tumor. As shown in Figure 3e, in the absence of esterase, the amount of PTX released from PTX-SM-TAR NPs in PBS at pH 7.4 was 35.52 ± 1.53% and pH 6.8 was 39.55 ± 2.59% after 48 h incubation. In the presence of esterase, the release rate of PTX in PBS at pH 7.4 was 52.77 ± 3.84% and pH 6.8 was 57.02 ± 0.51% after 48 h incubation.

### 2.5. Targeting Properties of PTX-SM-TAR NPs to NRP-1 In Vitro

In order to verify the targeting properties of PTX-SM-TAR NPs to NRP-1 in vitro, PTX-SM-TAR/C6 NPs were prepared by loading coumarin-6. PTX-SM-TAR/C6 NPs were co-cultured in 4T1-mCherry-Luc cells with high or low expression of NRP-1, followed by observing and quantifying cell uptake through LSCM and flow cytometry. The NRP-1 siRNA was used for establishing the 4T1-mCherry-Luc cell lines with NRP-1 down-regulated expression. A Western blot assay was used to determine the expression of the NRP-1 receptor. As depicted in Figure 4a, the expression of NRP-1 in the NRP-1 siRNA group was significantly lower than that in the control group, indicating that NRP-1 siRNA inhibited NRP-1 expression in 4T1-mCherry-Luc cell lines. As presented in Figure 4b,c, green fluorescence was visualized in the cytoplasm of cells after being cultured, and the fluorescence became stronger over time. At the same time point, the untreated cells with high expression of NRP-1 had more PTX-SM-TAR/C6 NPs uptake (Figure 4d). In order to further verify the targeting effect of PTX-SM-TAR NPs realized by the specific binding of TAR and NRP-1 receptor, an extra TAR peptide was added into cell culture to block the receptors. At the same time point, the uptake of PTX-SM-TAR/C6 NPs by wild 4T1-mCherry-Luc (NRP-1 higher expression) cells co-cultured with TAR peptide was significantly reduced compared with that without TAR peptide (Figure 4e). However, there was no significant difference in PTX-SM-TAR/C6 NPs uptake by NRP-1 low expressed cells with or without TAR peptide (Figure 4f). Moreover, the decrease in PTX-SM-TAR/C6 NPs uptake induced by TAR competition also occurred in the HUVEC cells with high NRP-1 expression (Appendix A).

### 2.6. Penetration of PTX-SM-TAR/C6 NPs in the Vascular Barrier In Vitro

In order to evaluate the penetration of PTX-SM-TAR NPs in tumor vessels, we measured the vascular barrier penetration efficiency of PTX-SM-TAR/C6 NPs in vitro. After the vascular barrier was successfully constructed in vitro (Figure 5a), PTX-SM-TAR/C6 NPs and coumarin-6 were added for incubation. The results of fluorescence determination were shown in Figure 5b,c. PTX-SM-TAR/C6 NPs showed a high penetration rate in the vascular barrier, which was significantly higher than that of the coumarin-6 group, showing a significant difference.

### 2.7. Transcellular Migration of PTX-SM-TAR/C6 NPs in 4T1-mCherry-Luc Cells

The transcellular migration experiment in 4T1-mCherry-Luc cells was conducted to evaluate the transcytosis of PTX-SM-TAR NPs. As shown in Figure 6, the coverslips (1)–(3) of the PTX-SM-TAR/C6 NPs group showed obvious fluorescence, indicating that PTX-SM-TAR/C6 NPs can be taken up by the cells in the coverslip (1) and can be transported to other coverslip cells. The coumarin-6 group also underwent efflux transport after uptake, but the intracellular fluorescence intensity was significantly weaker than that of the PTX-SM-TAR/C6 NPs group.

### 2.8. The Penetrating Efficiency of PTX-SM-TAR NPs in 3D Tumor Spheroids

A 3D tumor spheroid is a general model for evaluating permeability in vitro, which can simulate solid tumor tissue in vivo [54,55]. To evaluate the tumor penetrability of TAR peptide-mediated PTX-SM-TAR NPs, we used 4T1-mCherry-Luc cells to construct a multicellular spheroid (MCS) model, which was observed by CLSM after incubation. In this experiment, the tumor spheroids were incubated with PTX-SM-TAR/C6 NPs and coumarin-6, respectively, for 2 or 4 h and then scanned at an interval of 10 μm. As presented in Figure 7a,b, the fluorescence of the PTX-SM-TAR/C6 NPs treated group was distributed in most areas of MCS after incubation for 2 h while the fluorescence of the coumarin-6 treated group was mainly distributed in the outer edge of MCS. Over time, the fluorescence distribution of both groups developed more extensively. After incubation for 4 h, both groups could observe distinctive fluorescence in the internal region of the tumor spheroids (Figure 7c,d). Among them, the PTX-SM-TAR/C6 NPs treated group could observe obvious green fluorescence under different scanning layers while the fluorescence of the coumarin-6 treated group gradually disappeared after the *Z*-axis exceeded 60 μm.

### 2.9. Anti-Tubulin and Anti-Proliferation Effects on 4T1-mCherry-Luc Cells

To investigate the effects of PTX-SM-TAR NPs on microtubules in 4T1-mCherry-Luc cells, microtubules were observed by LSCM using an anti-tubulin antibody (Figure 8a) and the cytotoxicity effect of PTX-SM-TAR NPs against normal cells EA.hy926 (Appendix A). The microtubules in the control group were evenly distributed while the microtubules in the PTX and PTX-SM-TAR NPs treated groups aggregated and showed high green fluorescence. In addition, toxicological results against EA.hy926 cells showed that the linker SM had no toxicity to normal cells, and the PTX had slight toxicity to normal cells in the low-concentration. However, with the increase of drug concentration, the toxicity of PTX to normal cells also increased, and the conjugate PTX-SM-TAR showed the same trend as the PTX.

CCK-8 method was used to detect the anti-proliferation effect of PTX-SM-TAR NPs against 4T1-mCherry-Luc cells (Figure 8b). From 0.0001 nM to 10,000 nM, the inhibition rate of TAR, PTX, TAR plus PTX, and PTX-SM-TAR NPs on 4T1-mCherry-Luc cells increased gradually. At all concentrations, the cell proliferation inhibition rate of the PTX-SM-TAR NPs group was higher than that of the PTX group with statistical differences, which may be caused by an increased cell uptake of PTX-SM-TAR NPs mediated by TAR peptide.

### 2.10. PTX-SM-TAR NPs Inhibit Migration on 4T1-mCherry-Luc Cells

To assess the migration inhibitory effect of PTX-SM-TAR NPs on 4T1-mCherry-Luc cells, the wound healing assay was conducted. At 24 h, the migration rate was 36.89 ± 2.09% in the control group, 32.23 ± 3.81% and 32.53 ± 1.43% in 10 nM PTX-SM-TAR NPs and PTX groups, and 20.10 ± 3.15% and 18.96 ± 2.76% in 100 nM PTX-SM-TAR NPs and PTX groups (Figure 8c,d). The migration rate of 4T1-mCherry-Luc cells treated with PTX-SM-TAR NPs was equivalent to that of PTX.

### 2.11. Evaluation of Antitumor Efficacy In Vivo

The mice TNBC 4T1-mCherry-Luc model was constructed by subcutaneous inoculation to study the anti-tumor effect of PTX-SM-TAR NPs in vivo. 4T1-mCherry-Luc cells reacted with D-luciferin potassium salt to produce bioluminescence, which was determined by an IVIS spectrum imaging system. The intensity of the fluorescence signal is related to tumor size, hence tumor growth in mice can be monitored in real-time. As shown in Figure 9a, the tumor growth rate of the PTX-SM-TAR NPs group was lower than that of the NS, TAR, and PTX groups. At the end of the experiment, the tumor tissue was weighed, and the results were consistent with the trend of fluorescence intensity (Figure 9b). The tumor inhibition rate was 43.24% in the PTX-SM-TAR NPs, 28.47% in the PTX, and 7.81% in the TAR. The tumor inhibitory effect of the PTX-SM-TAR NPs group was stronger than that of the PTX group, and the difference was significant.

In addition, the TUNEL and H&E staining results of tumor tissues showed that the TNBC treated with PTX and PTX-SM-TAR NPs had obvious apoptosis and serious tissue damage (Figure 10). On the contrary, NS group tumor cells were arranged closely and orderly, and no visible apoptosis or necrosis was observed. The changes in mice weight during treatment were measured to evaluate the toxic and side effects of PTX-SM-TAR NPs (Figure 9c). The body weight of mice in TAR, PTX, and PTX-SM-TAR NPs groups remained stable during the experiment while the body weight of mice in the NS group began to decrease gradually after 8 days of administration.

## 3. Discussion

The TAR peptide combines the advantages of A7R and TAT peptide, which not only specifically target NRP-1 on cells, but also can efficiently transport across the cell membrane. In order to verify whether the TAR peptide carrier can successfully and effectively transport drugs to the tumor site, the fluorescence distribution of Cy5.5-MAL-TAR in vivo was monitored by an IVIS spectrum imaging system. In vivo imaging data showed that Cy5.5-MAL-TAR remarkably increased the content of Cy5.5-MAL and prolonged its retention time in the tumor site. It was observed with staining that Cy5.5-MAL-TAR was widely dispersed in tumor tissues, covering all micro-vessels and tumor cells, while free Cy5.5-MAL only diverged in a spot-like manner, indicating that TAR acted as a carrier and delivered a large amount of Cy5.5-MAL into tumor tissues. In addition, the TAR vector targets more Cy5.5-MAL to the tumor site, thereby reducing the distribution of Cy5.5-MAL in normal tissues, which may facilitate the reduction of systemic toxicity in the future delivery of drugs.

Because of its excellent delivery performance, the TAR peptide is used to modify PTX to improve its deficiency. Through structural analysis, it was found that the lysine and arginine in the TAR peptide played an important role in cell penetration. In addition, arginine at the C-terminal was the guarantee for the realization of targeting. When a TAR peptide was used for the structural modification of PTX, it needed to be ensured that the active sites of TAR peptide were not covered. Therefore, a cysteine was added to the N-terminal of TAR to form a thioether bond, which is common in ADCs design, to finally synthesize the intelligent PDC. In the structure of PTX, the C2′-OH is not only the active site of PTX, but also the commonly used modification site, which can be temporarily shielded to synthesize inactive prodrugs. Therefore, SA-MAL was used as the connecting arm to connect TAR with PTX through a three-step reaction, forming the acid- and esterase-sensitive prodrug PTX-SM-TAR.

PTX-SM-TAR is an amphiphilic molecule with self-assembly properties and can be soluble in an aqueous solution. We hypothesize that the Π-Π interaction between PTX molecules and the hydrogen bonding between TAR molecules may lead to the formation of a hydrophilic peptide shell, and the hydrophobic PTX was wrapped in the core to form water-soluble micelles. An act verified by TEM and DLS experiments, when the concentration was higher than 3.95 × 10^−6^ mol/L, PTX-SM-TAR could self-assemble to form uniform and dispersed spherical nanoparticles with particles size of 94.70 ± 1.87 nm. The above data showed that PTX-SM-TAR molecules could increase the water solubility of PTX through self-assembly into nanoparticles.

To verify the targeting properties of PTX-SM-TAR NPs, in vitro cellular uptake was performed in 4T1-mCherry-Luc cells with high and low expression of NRP-1. After being co-cultured with PTX-SM-TAR/C6 NPs solution, untreated 4T1-mCherry-Luc cells showed stronger fluorescence than NRP-1 siRNA treated 4T1-mCherry-Luc cells. Assuming that this phenomenon is caused by receptor-mediated endocytosis, we further conducted the competitive experiment. After TAR incubation, the uptake of fluorescent substances in normal 4T1-mCherry-Luc cells was significantly decreased with TAR competition while that in NRP-1 siRNA treated cells was not significantly changed. TAR indeed has the ability to block the entry of PTX-SM-TAR/C6 NPs into cells with high NRP-1 expression, indicating that PTX-SM-TAR/C6 NPs have the same mechanism as TAR for cell entry. Therefore, it can be inferred that PTX-SM-TAR has the targeting ability to NRP-1 and enter into cells through receptor-mediated endocytosis, which is helpful for PTX to target new blood vessels and tumor sites where NRP-1 is highly expressed.

Tumor therapeutic drugs should not only actively target the tumor but also effectively penetrate the tumor tissue after reaching the tumor site. In this study, we first measured the permeability of PTX-SM-TAR NPs in the vascular barrier. The results showed that PTX-SM-TAR NPs could successfully cross the vascular barrier, which facilitates drug penetration from blood vessels into the tumor solid site to increase drug concentration at the tumor site. Then, the transcellular migration experiment showed that PTX-SM-TAR NPs can be transported between cells, which facilitates drug delivery from tumor edge cells to interior cells. In addition, we utilized the 4T1-mCherry-Luc cells tumor spheroid to simulate the penetration of PTX-SM-TAR NPs in vivo. Compared with free coumarin-6, coumarin-6 encapsulated in PTX-SM-TAR NPs could penetrate deeper into the tumor spheroid, and the depth was increased over time. TAR peptide enhanced the penetration of PTX, which contributed to its effects on cells within the tumor parenchyma.

TAR reacted with the active site 2′-OH of PTX to form the prodrug PTX-SM-TAR NPs. Prodrugs are generally inactive during normal circulation in vivo and take effect in abnormal tumor microenvironments by stimulating the release of free active drugs. The ester bond is considered to be an acid- and enzyme-sensitive bond. When the pH decreases or the content of esterase increases, the unstable ester bond will be disconnected. In the release experiment, PTX can be released slowly under different conditions without burst release. When the conditions of low pH and high esterase were satisfied at the same time (tumor simulation environment), the release of PTX reached the maximum. The results demonstrated that the release of PTX in PTX-SM-TAR NPs prodrug was controlled and sustained.

In vitro microtubule experiment, compared with the control group, 4T1-mCherry-Luc cells treated with PTX and PTX-SM-TAR NPs showed a dynamic imbalance of microtubules in the cytoplasm, which was consistent with the mechanism of PTX stabilizing microtubule polymerization. In vitro cell anti-proliferation experiments showed that the growth of 4T1-mCherry-Luc cells in each group was inhibited in a dose-dependent manner. Among them, the conjugate PTX-SM-TAR NPs exhibited stronger cytotoxicity against 4T1-mCherry-Luc cells than free PTX. These results indicated that PTX modified by TAR peptide did not affect its biological activity, and 4T1-mCherry-Luc cells took up more PTX-SM-TAR mediated by TAR, resulting in increased intracellular content of PTX. In vitro cytotoxicity assay against normal cells EA.hy926 showed that the linker SM is low toxicity, but PTX and PTX-SM-TAR NPs were not completely safe and non-toxic to normal cells. It is inferred that the toxicity of PTX-SM-TAR to normal cells is mainly attributed to PTX, and the linker we used is safe. In vivo, the PTX-SM-TAR NPs mainly tend to tumors and accumulate less in other normal tissues, thereby reducing the damage to normal tissue cells. Moreover, PTX-SM-TAR NPs and PTX also can effectively inhibit the migration of 4T1-mCherry-Luc cells.

The antitumor effect of PTX-SM-TAR NPs in vivo was evaluated by 4T1-mCherry-Luc animal models. The distribution of PTX-SM-TAR NPs in the tumor leads to distinct apoptosis and tissue damage. PTX-SM-TAR NPs can effectively inhibit the growth of TNBC tumors, which is superior to free PTX. In summary, TAR peptide-mediated PTX-SM-TAR NPs can actively target and penetrate into tumor tissue in vivo, and effectively inhibit tumor growth.

## 4. Materials and Methods

### 4.1. Materials

The paclitaxel was purchased from Wuxi Taxus Pharmaceutical Co., Ltd. (Jiangsu, China). Succinic anhydride (SA), 4-dimethyl aminopyridine (DMAP), 1-hydroxy-benzotriazole (HoBt), and 2-(1H-Benzotriazole-1-yl)-1,1,3,3-tetramethyluronium hexafluorophosphate (HBTU) were purchased from Macklin (Shanghai, China). *N*,*N*-Diisopropylethylamine (DIPEA) was purchased from Aladdin (Shanghai, China). *N*-(2-Aminoethyl) maleimide trifluoroacetate salt (MAL) was purchased from J&K Chemical Ltd. (Beijing, China). TAR peptide (CRKKRRQRRRATWLPPR) was synthesized from China Peptide Co., Ltd. (Shanghai, China). Esterase and coumarin-6 were purchased from Shanghai yuanye Bio-Technology Co., Ltd. (Shanghai, China) Rabbit anti-NRP-1 antibody was purchased from Abcam (Cambridge, UK). Rabbit anti-β-actin antibody and secondary antibody (anti-rabbit) were purchased from ProteinTech (Beijing, China). 3-(4,5-dimethylthiazol-2-yl)-2,5-diphenyltetrazolium bromide (MTT) was obtained from BioFroxx (Einhausen, Germany). Cy5.5 maleimide (non-sulfonated, Cy5.5-MAL) was obtained from APExBIO (Houston, TX, USA). Tubulin-tracker green and DAPI were obtained from Beyotime (Shanghai, China). Regular agarose G-10 was purchased from Biowest (Spain). F-12K medium, heparin and endothelial cell growth supplement (ECGS) were obtained from MacGene (Beijing, China). 1640 medium was obtained from Meilunbio (Dalian, China).

### 4.2. Cell lines and Cell Culture

Human umbilical vein endothelial cells (HUVEC) were obtained from ATCC (Manassas, VA, USA), and 4T1-mCherry-Luc cells (TNBC cells) were obtained from Caliper Life Sciences (Boston, MA, USA). HUVEC cells were cultured in F12K medium containing 10% fetal calf serum (FBS), 1% ECGs, and 1% heparin. 4T1-mCherry-Luc cells were cultured in 1640 medium containing 10% FBS. Both of them were incubated at 37 °C with 5% CO_2_.

### 4.3. Animal Model

Healthy female BALB/c mice (18–20 g) were purchased from Shandong University Laboratory Animal Center. Mice were housed in a pathogen-free environment under conditions of 25 ± 2 °C, 50 ± 5% relative humidity, and 12-h light/dark cycles. They were provided with food and water ad libitum. All animal experiment protocols were approved by the Ethics Committee of School of Pharmaceutical Sciences, Shandong University (22017) and implemented according to the regulations of the Shandong Council on Animal Care. To establish the mice TNBC model, 1.0 × 10^6^ 4T1-mCherry-Luc cells were subcutaneously injected into the left forelimb armpit of mice. D-fluorescein potassium salt solution was injected into mice through the abdominal and reacted with 4T1-mCherry-Luc cells to produce bioluminescence. Then, the fluorescence signals were observed and counted by an IVIS Spectrum imaging system (IVIS Kinetic).

### 4.4. Validation of the Targeting and Drug Delivery Properties of TAR Peptide

The surface plasmon resonance (SPR) detection technique was used to determine the binding affinity of TAR peptide to NRP-1 by Biacore T200 instrument (GE). First, the purified NRP-1 protein was immobilized on the CM5 chip. Then, using HBS-EP as a running buffer, a gradient-diluted TAR peptide solution was configured to flow through the surface of the chip for binding and dissociation. Finally, the data were analyzed using the Biacore T200 evaluation software 2.0.1.

Cy5.5-MAL-TAR was synthesized by covalently linking the near-infrared dye Cy5.5-MAL with TAR peptide. The in vivo distribution experiment was carried out to determine the in vivo delivery capacity of TAR peptide to drugs. The 4T1-mCherry-Luc tumor-bearing BALB/c mice were treated with Cy5.5-MAL-TAR and Cy5.5-MAL (4 mg/kg) via tail vein injection (*n* = 3). Three mice in each group, a total of six mice were used. At 2, 4, 8, 12, and 24 h after treatment, mice of each group were taken to capture the fluorescent imaging of bio-distribution. At 8 h after administration, mice were sacrificed, and their major organs were extracted for future in vitro imaging.

### 4.5. Synthesis of PTX-SA

PTX (427.5 mg, 0.5 mmol), SA (32.5 mg, 0.3 mmol), and DMAP (5 mg, 0.03 mmol) were dissolved in 10 mL dichloromethane. The reaction solution was stirred for 6 h at room temperature in dark, and real-time monitoring was carried out with thin-layer chromatography (TLC). After the reaction, saturated NaHCO_3_ and saturated NaCl solutions were used for extraction, then the products were purified by silica gel column (methanol/dichloromethane). Finally, PTX-SA was obtained by rotary evaporation (80% yield) and analyzed by ^1^H-NMR (AVANCE III 600) and Q-TOF HRMS (Bruker, Maxis II). ^1^H-NMR (DMSO-d_6_, 600 MHz): δppm 0.991 (s, 3H), 1.021 (s, 3H), 1.470–1.511 (m, 4H), 1.607–1.649 (m, 1H), 1.753 (s, 3H), 1.773–1.814 (m, 1H), 2.108 (s, 3H), 2.240 (s, 3H), 2.295–2.346 (m, 1H), 2.612–2.633 (m, 2H), 3.429–3.449 (m, 2H), 3.569–3.587 (m, 1H), 3.984–4.021 (m, 2H), 4.085–4.126 (m, 1H), 4.899–4.949 (m, 2H), 5.341–5.356 (d, 1H), 5.401–5.413 (d, 1H), 5.513–5.542 (t, 1H), 6.292 (s, 1H), 7.174–7.203 (m, 1H), 7.429–7.467 (m, 4H), 7.488–7.512 (m, 2H), 7.555–7.580 (m, 1H), 7.655–7.680 (m, 2H), 7.727–7.453 (m, 1H), 7.839–7.853 (m, 2H), 7.971–7.985 (m, 2H), 9.241–9.255 (d, 1H), 12.257 (s, 1H). The calculated molecular weight for C_51_H_55_NO_17_ is 953.3470. [M+Na]^+^ HRMS *m*/*z*: 976.3346 (Appendix A).

### 4.6. Synthesis of PTX-SA-MAL

PTX-SA (40 mg, 0.042 mmol) was dissolved in 2 mL *N*,*N*-Dimethylformamide (DMF), then MAL (11.2 mg, 0.044 mmol), HoBt (6.24 mg, 0.046 mmol), HBTU (17.48 mg, 0.046 mmol), and DIPEA (27.72 mg, 0.168 mmol) were added to the solution and stirred at room temperature overnight. DMF was first removed with rotary evaporation, and then the products were redissolved with dichloromethane (DCM) and extracted three times with saturated NaCl solution. Finally, PTX-SA-MAL was purified with column chromatography (85% yield) and characterized by ^1^H-NMR and Q-TOF HRMS. ^1^H-NMR (DMSO-d_6_, 600 MHz): δppm 0.997 (s, 3H), 1.025 (s, 3H), 1.484–1.510 (m, 4H), 1.601–1.655 (m, 1H), 1.766 (s, 3H), 1.777–1.818 (m, 1H), 2.101 (s, 3H), 2.233 (s, 3H), 2.254–2.344 (m, 3H), 2.556–2.617 (m, 2H), 3.086–3.195 (m, 2H), 3.392–3.441 (m, 2H), 3.569–3.581 (d, 1H), 3.989–4.0442 (m, 2H), 4.081–4.123 (m, 1H), 4.905–4.917 (d, 2H), 5.330–5.345 (d, 1H), 5.408–5.420 (d, 1H), 5.517–5.546 (t, 1H), 6.289 (s, 1H), 6.988 (s, 2H), 7.174–7.212 (m, 1H), 7.431–7.463 (m, 4H), 7.483–7.508 (t, 2H), 7.549–7.573 (t, 1H), 7.651–7.676 (t, 2H), 7.724–7.749 (t, 1H), 7.846–7.859 (d, 2H), 7.976–8.005 (m, 3H), 9.202–9.216 (d, 1H). The calculated molecular weight for C_57_H_61_N_3_O_18_ is 1075.3950. [M+H]^+^ HRMS m/z: 1076.4059, [M+NH4]^+^ HRMS m/z: 1093.4322 (Appendix A).

### 4.7. Synthesis of PTX-SM-TAR

TAR peptide (90 mg, 0.040 mmol) was dissolved in 2 mL PBS solution at pH 7.4, and PTX-SA-MAL (60 mg, 0.05 mmol) was dissolved in 3 mL methanol. The PTX-SA-MAL solution was slowly dropped into the TAR peptide solution. The reaction mixtures were stirred at room temperature in dark for 2 h, monitored with analytical HPLC (Agilent 1200 Infinity II). The final product PTX-SM-TAR (SM is the abbreviation of SA-MAL) was purified with semi-preparative HPLC (H&E), and its structure was verified by ^1^H-NMR and Q-TOF HRMS (75% yield, 10–100% acetonitrile containing 0.1% trifluoroacetic acid). The calculated molecular weight for C_153_H_231_N_45_O_38_S is 3340.8680. [M+2H]^2+^ HRMS *m*/*z*: 1671.3668. [M+3H]^3+^ HRMS *m*/*z*: 1114.5837 (Figure 2).

### 4.8. Determination of Critical Aggregation Concentration (CAC)

A series of PTX-SM-TAR solutions were prepared using sterile water, following the concentration range from 2.34 × 10^−8^ mol/L to 1.90 × 10^−4^ mol/L. Subsequently, CAC was determined by the digital conductivity meter (DDS-11A) at 25 °C.

### 4.9. Characterization of PTX-SM-TAR Nanoparticles (NPs)

The morphology of PTX-SM-TAR NPs was characterized by transmission electron microscopy (TEM, JEM-1200EX). The particle size distribution and zeta potential of PTX-SM-TAR NPs were measured by dynamic light scattering (DLS, BIC-Brookhaven, New York, NY, USA).

### 4.10. In Vitro Release Study for PTX-SM-TAR NPs

The acid and esterase sensitivity of PTX-SM-TAR NPs was ascertained by in vitro release experiment, and PBS solution containing 1% Tween 80 was used as the release medium. There are four different release conditions, pH 7.4, pH 6.8, pH 7.4 plus 100 U/L esterase, and pH 6.8 plus 100 U/L esterase were set to simulate the physiological environment. An amount of 1 mL PTX-SM-TAR NPs solution with a concentration of 1.2 mg/mL was prepared and packed into a dialysis bag (MWCO = 1000 Da). These bags were placed in a container containing 100 mL release medium and incubated in a shaker (ZQZY-85CN) at 37 °C, 100 r/min. At the specific time point, 1 mL medium was removed from the container, and then 1 mL fresh medium was added. The content of PTX in all samples was measured by analytical HPLC (10–100% acetonitrile over 0–15 min).

### 4.11. Transfection of siRNA into 4T1-mCherry-Luc Cells

4T1-mCherry-Luc cells with high and low expression of NRP-1 were used to detect the targeting of PTX-SM-TAR to NRP-1. NRP-1 siRNA was used to construct 4T1-mCherry-Luc cells with NRP-1 down-regulated expression. NRP1-Mus-1755 siRNA was designed and synthesized by GenePharma (Suzhou, China). The transfection of siRNA was performed using the Invitrogen lipofectamine RNA iMAX transfection reagent in accordance with the manufacturer’s instructions. An amount of 5 × 10^5^ 4T1-mCherry-Luc cells were planted into the 6-well plate and cultured overnight. Then, siRNA (25 pmol) was mixed with transfection reagent at room temperature for 5 min and then added to the plate for 6 h.

### 4.12. Western Blotting Analysis

Cells were lysed in a mixture of the protease inhibitor, phosphatase inhibitor, and RIPA lysate for 30 min, and the total proteins were extracted after centrifugation. The concentration of extracted proteins was determined by the BCA kit. The total proteins extracted from cells were separated by SDS-PAGE and transferred to a PVDF membrane. The membrane was blocked in 5% skim milk powder solution for 2 h and then was incubated with primary antibody at 4 °C overnight and with secondary antibody at room temperature for 1 h. Finally, proteins were colored using enhanced chemiluminescence (ECL) and imaged on a chemiluminescence gel imaging analysis system (ChemiDoc XRS+).

### 4.13. Cellular Uptake of PTX-SM-TAR/C6 NPs

Coumarin-6-loaded PTX-SM-TAR NPs were prepared and applied to explore cell uptake. An amount of 6.5 × 10^4^ NRP-1 siRNA treated and untreated 4T1-mCherry-Luc cells were seeded in a glass-bottom cell culture dish and cultured overnight. After cell adherence, 100 ng/mL coumarin-6 equivalent PTX-SM-TAR/C6 NPs solution was added to each group, and 30 μM TAR peptide was extra added to co-incubation in the competition inhibition groups. After incubation for 0.5 and 1 h, the cells were washed with PBS, fixed with 4% paraformaldehyde for 30 min, and then stained with DAPI dye for 5 min. The laser scanning confocal microscope (LSCM, Dragonfly 200) was used for photographing.

For flow cytometry analysis, cells were planted in 6-well plates with a density of 3 × 10^5^ cells per well. After incubation overnight, PTX-SM-TAR/C6 NPs solution with or without 30 μM TAR peptide was added and incubated for 0.5 and 1 h. After washing the cells with PBS, 500 μL trypsin was added to each well for 1 min, and then, the cells were collected and centrifuged at 1000 r/min for 5 min. The supernatant was discarded and cells were resuspended with 300–500 μL PBS. The intracellular fluorescence was quantified by a flow cytometer (Accuri C6 Plus).

### 4.14. Penetration of PTX-SM-TAR/C6 NPs in the Vascular Barrier In Vitro

The tightly connected EA.hy926 cell layer was used to simulate the tumor vascular barrier in vivo. EA.hy926 cells were seeded at a density of 1 × 10^6^ per well in a 0.44 μm Transwell upper chamber for one week. When the transmembrane resistance TEER reaches 200 Ω detected by the resistance detector, it was considered that the in vitro vascular barrier was successfully constructed. An amount of 250 ng/mL coumarin-6 equivalent PTX-SM-TAR/C6 NPs solution was added to the Transwell upper chamber, and PBS solution was added to the lower chamber. At 2, 4, 8, and 12 h, the lower chamber solution was taken for fluorescence determination. The fluorescence intensity of the initial addition of coumarin-6 was 100%, and the relative percentage of the fluorescence intensity of the lower chamber was calculated as the penetration rate of the vascular barrier. After incubation for 12 h, the Transwell chambers were washed with PBS, fixed with 4% paraformaldehyde for 30 min, and observed by LSCM.

### 4.15. Transcellular Migration of PTX-SM-TAR/C6 NPs in 4T1-mCherry-Luc Cells

An amount of 5 × 10^4^ 4T1-mCherry-Luc cells were seeded in coverslips (1)–(3) and cultured overnight. After cell adherence, 100 ng/mL coumarin-6 equivalent PTX-SM-TAR/C6 NPs solution was added to the coverslip (1) for 4h at first. Then, the cells on the coverslip (1) were washed with PBS three times and co-incubated with cells on the coverslip (2) in fresh medium for 4h. As mentioned above, cells on the coverslip (2) and coverslip (3) were co-incubated for 4h. Finally, the cells were washed with PBS, fixed with 4% paraformaldehyde for 30 min, and then stained with DAPI dye for 5 min before being imaged with LSCM.

### 4.16. Penetration of PTX-SM-TAR/C6 NPs in 3D Tumor Spheroids

After adding 150–200 μL of 2% agarose solution to each well, the 48-well plate was placed on the ultra-clean bench under UV irradiation for 30 min before solidification. The 4T1-mCherry-Luc cells were planted into the plate at the density of 4 × 10^4^ cells per well and cultured in the incubator for one week. After the cells aggregated into spherical shapes, tumor spheroids were transferred to confocal dishes and incubated with PTX-SM-TAR/C6 NPs and coumarin-6 solution for 2 and 4 h. The tumor spheroids were washed with PBS, fixed with 4% paraformaldehyde for 30 min, and observed with LSCM.

### 4.17. Anti-Tubulin Effects of PTX-SM-TAR NPs

4T1-mCherry-Luc cells were cultured overnight in 6-well plates and treated with PTX and PTX-SM-TAR NPs (31.25 nM), respectively, for 24 h. After that, the cells were washed with PBS and fixed with 4% paraformaldehyde for 20 min. Subsequently, 2% BSA was used for blocking, and 0.1% Triton X-100 was used for permeation. Finally, Tubulin-tracker green staining solution was used for 45 min, and LSCM was used for photography observation.

### 4.18. Anti-Proliferative Activity of PTX-SM-TAR NPs

The anti-proliferative activity of PTX-SM-TAR NPs on 4T1-mCherry-Luc cells was evaluated by CCK-8 methods. Briefly, the 4T1-mCherry-Luc cells in the logarithmic phase were digested and spread into the 96-well plate at the density of 8 × 10^3^ cells per well. Cells were cultured overnight and then incubated with 200 μL different concentrations of drugs. After incubation for 48 h, the drug-containing medium was discarded and 100 μL fresh basic medium was added per well, followed by 10 μL CCK8 for 1–4 h. The cell viability was measured at 450 nm by a microplate reader (BIO-RAD, Hercules, CA, USA).

### 4.19. Migration Assay of PTX-SM-TAR NPs

The effect of PTX-SM-TAR NPs and PTX on 4T1-mCherry-Luc cell migration was evaluated with wound-healing assay. 4T1-mCherry-Luc cells were seeded in 6-well plates at a density of 3 × 10^5^ cells per well. After overnight culture, the cells were wounded linearly by a 200 µL pipette tip. Then, cells were incubated in the medium containing PTX-SM-TAR NPs and PTX (100 nM). Images of 0 and 24 h were captured by an inverted microscope.

### 4.20. In Vivo Tumor Evaluation Studies

Four days after tumor cell implantation, mice were randomly divided into four groups (*n* = 7). Seven mice in each group, a total of twenty-eight mice were used. Then, saline, TAR peptide (10.6 mg/kg), PTX (4 mg/kg), and PTX-SM-TAR NPs (15.6 mg/kg) were injected through the tail vein every two days. Tumor size and body weight were measured every 4 d. After 16 d of treatment, mice were sacrificed and tumors were removed, weighed, and further analyzed.

### 4.21. Histology Analysis

Mice tumor tissues were fixed in 4% paraformaldehyde, decolored and embedded in paraffin, and sliced about 3–5 microns in thickness. Tumor paraffin sections were stained with H&E and TUNEL, respectively, and scanned by a digital slice scanning microscope (VS120).

### 4.22. Statistical Analysis

Statistical analyses were performed with GraphPad Prism 8.0. The Student’s *t*-test or one-way ANOVA was conducted to identify the differences between groups. A value of *p* < 0.05 was considered statistically significant.

## 5. Conclusions

In this study, we designed and synthesized a transcytosable tumor-targeting prodrug delivery system based on the concept of “PDCs”, which covalently bonded a bifunctional TAR peptide and chemotherapy drug PTX. The final product PTX-SM-TAR could self-assemble into nanoparticles with a shell-core structure driven by hydrophilic TAR peptide and hydrophobic PTX, and it greatly improved the water solubility of PTX. In addition, with the mediation of the TAR peptide, PTX could be specifically delivered to the tumor site. The transvascular transport, intracellular delivery, and tumor penetration of PTX were increased as well, which leads to the highly effective extravasation on vascular, transcytosis between tumor cells, and infiltration in tumors. Afterward, due to the sensitivity of the ester bond, the prodrug PTX-SM-TAR can release free PTX in the abnormal tumor microenvironment for action, expecting to reduce systemic toxicity. Finally, combined with the above advantages, PTX-SM-TAR NPs could effectively inhibit the growth of TNBC tumors and had a stronger effect in promoting apoptosis and inhibiting tumor growth than PTX. In conclusion, PTX-SM-TAR NPs may present a new strategy for a targeted therapy for TNBC.

## Figures and Tables

**Figure 1 ijms-24-04646-f001:**
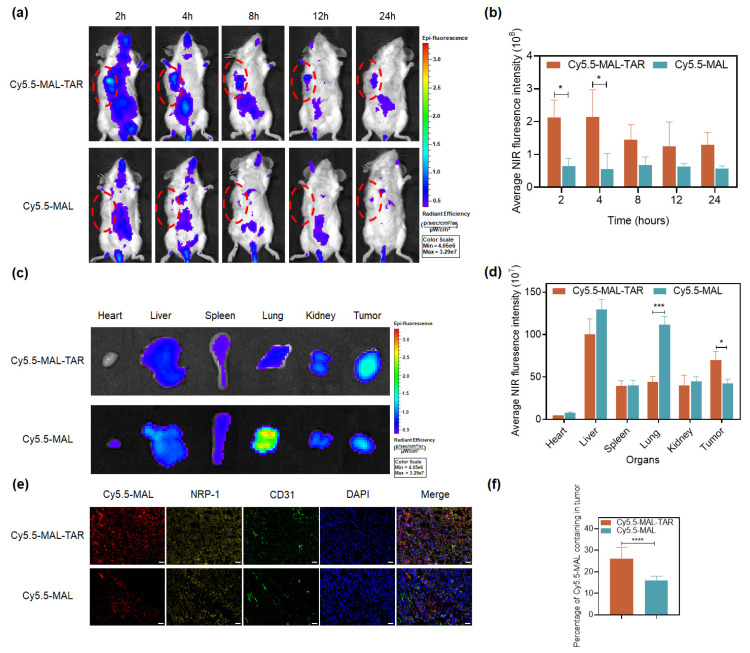
Tumor targeting delivery ability of TAR peptide. (**a**,**b**) In vivo fluorescence images and quantitative analysis of fluorescence intensity in mice after the treatment of Cy5.5-MAL-TAR and Cy5.5-MAL at different time points (mean ± SEM, *n* = 3, * indicate *p* < 0.05). (**c**,**d**) Ex vivo fluorescence images and quantitative analysis of fluorescence intensity in major organs and tumors after the treatment of Cy5.5-MAL-TAR and Cy5.5-MAL for 8 h (mean ± SEM, *n* = 3, * and *** indicate *p* < 0.05 and *p* < 0.001). (**e**) Distribution of Cy5.5-MAL-TAR and Cy5.5-MAL in tumor tissues of mice. Tumor vessels were labeled with CD31 antibody (green), the NRP-1 proteins were stained with NRP-1 antibody (yellow), and the nuclei were stained with DAPI (blue) and Cy5.5-MAL (red). The scale bar is 100 μm. (**f**) Statistical percentage of Cy5.5-MAL in tumor tissues (*n* = 10, **** indicate *p* < 0.0001).

**Figure 2 ijms-24-04646-f002:**
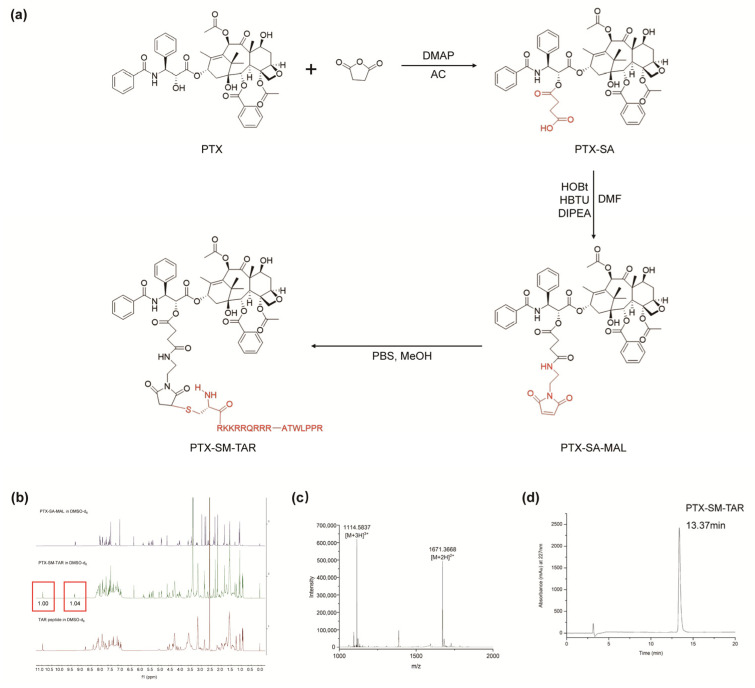
Synthesis and characterization of PTX-SM-TAR. (**a**) Synthetic route of PTX-SM-TAR. (**b**–**d**) ^1^H-NMR, Q-TOF HRMS, and HPLC spectrum of PTX-SM-TAR.

**Figure 3 ijms-24-04646-f003:**
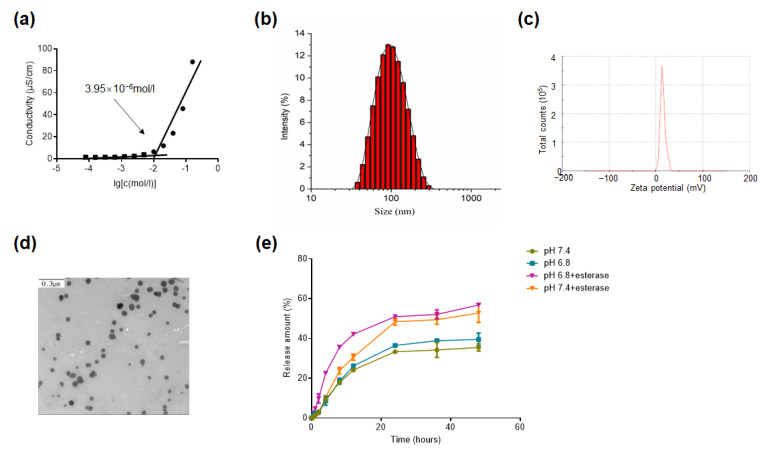
Representation of nanoparticles. (**a**) Determination of critical aggregation concentration of PTX-SM-TAR solution by conductivity method. (**b**) Size distribution of PTX-SM-TAR NPs. (**c**) Zeta potential of PTX-SM-TAR NPs. (**d**) TEM image of PTX-SM-TAR NPs. (**e**) In vitro release curves of PTX-SM-TAR NPs in different conditions (pH 6.8, pH 7.4, pH 6.8 with esterase, and pH 7.4 with esterase).

**Figure 4 ijms-24-04646-f004:**
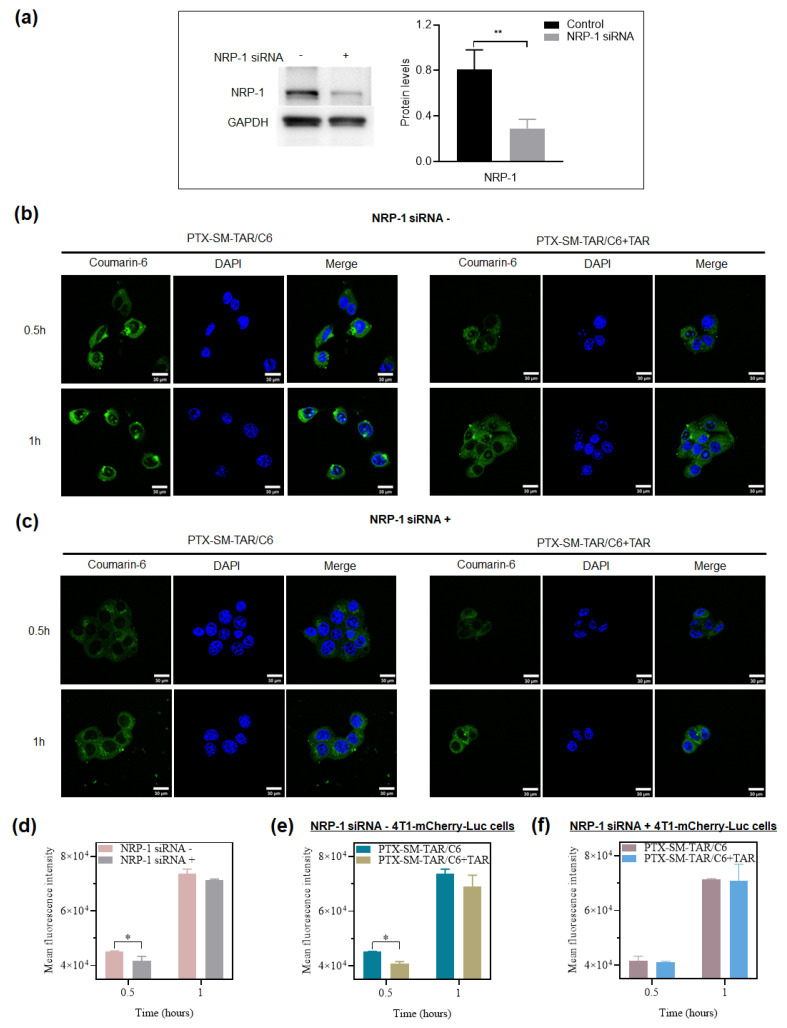
The targeting properties of PTX-SM-TAR NPs. (**a**) Western blot analysis of NRP-1 expression in NRP-1 siRNA untreated and treated 4T1-mCherry-Luc cells (mean ± SEM, *n* = 3, ** indicates *p* < 0.01). (**b**) Confocal images of untreated 4T1-mCherry-Luc cells (NRP-1 siRNA-, NRP1 high expressed). The cell nuclei were stained with DAPI (blue), and the PTX-SM-TAR NPs were dyed with coumarin-6 (green). The scale bar is 30 μm. (**c**) Confocal images of NRP-1 siRNA treated 4T1-mCherry-Luc cells (NRP-1 siRNA+, NRP1 low expressed). The cell nuclei were stained with DAPI (blue), and the PTX-SM-TAR NPs were dyed with coumarin-6 (green). The scale bar is 30 μm. (**d**) Flow cytometry analysis of NRP-1 siRNA untreated and treated 4T1-mCherry-Luc cells. (mean ± SEM, *n* = 3, * indicates *p* < 0.05). (**e**) Flow cytometry analysis of wild 4T1-mCherry-Luc cells treated with PTX-SM-TAR/C6 NPs and PTX-SM-TAR/C6 NPs plus TAR peptide groups (mean ± SEM, *n* = 3, * indicates *p* < 0.05). (**f**) Flow cytometry analysis of NRP-1 siRNA treated 4T1-mCherry-Luc cells treated with PTX-SM-TAR/C6 NPs and PTX-SM-TAR/C6 NPs plus TAR peptide groups (*n* = 3).

**Figure 5 ijms-24-04646-f005:**
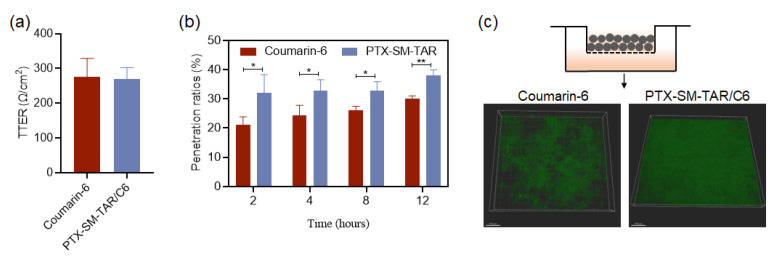
Penetration of PTX-SM-TAR/C6 NPs in the vascular barrier in vitro. (**a**) Barrier resistance value in vitro (mean ± SEM, *n* = 3). (**b**) Penetrating capacity of PTX-SM-TAR/C6 NPs and coumarin-6 in the lower chamber of the vascular barrier (mean ± SEM, *n* = 3, * indicates *p* < 0.05, and ** indicates *p* < 0.01). (**c**) 3D-images of vascular barrier treated with PTX-SM-TAR/C6 NPs and coumarin-6 for 12 h. Green, coumarin-6. The scale bar is 150 μm,.

**Figure 6 ijms-24-04646-f006:**
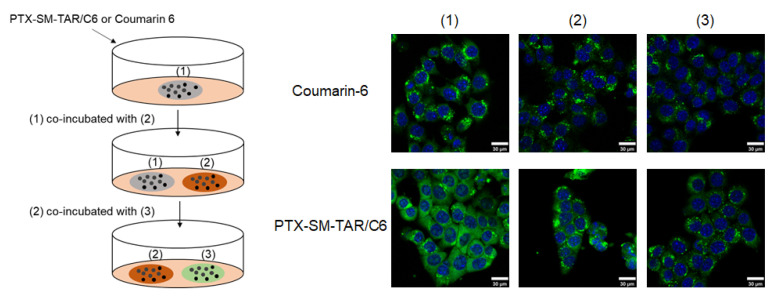
Intracellular migration of PTX-SM-TAR/C6 NPs and coumarin-6 visualized by LSCM. The cell nuclei were stained with DAPI (blue), and the PTX-SM-TAR NPs were dyed with coumarin-6 (green). The scale bar is 30 μm.

**Figure 7 ijms-24-04646-f007:**
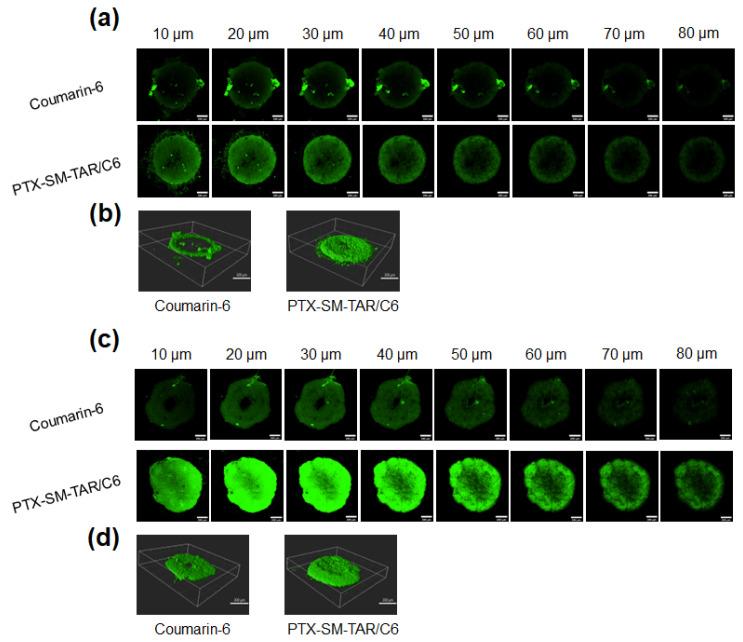
Penetration of PTX-SM-TAR NPs in 4T1-mCherry-Luc 3D tumor spheroids. (**a**,**c**) Scanned images of tumor spheroids treated with PTX-SM-TAR/C6 NPs and coumarin-6 for 2 and 4 h, respectively. Green, coumarin-6. The scale bar is 200 μm, the interval is 10 μm. (**b**,**d**) 3D-reconstruction of tumor spheroids in PTX-SM-TAR/C6 NPs and coumarin-6 for 2 and 4 h, respectively. The scale bar is 300 μm.

**Figure 8 ijms-24-04646-f008:**
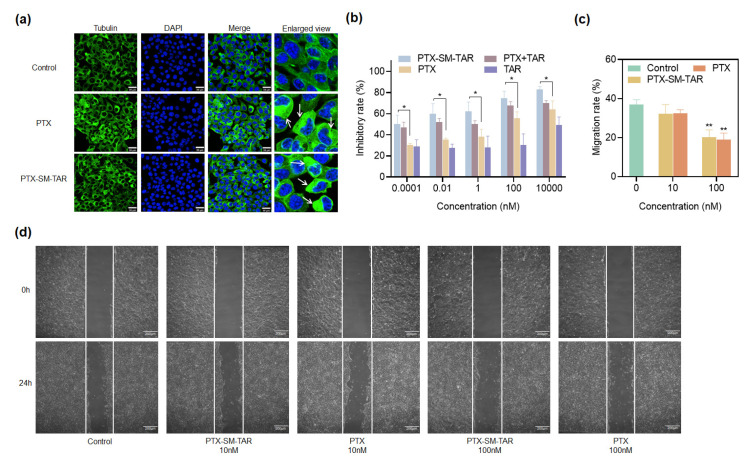
In vitro activity of PTX-SM-TAR NPs on 4T1-mCherry-Luc cells. (**a**) Confocal microscopy images of 4T1-mCherry-Luc cells after incubation with PTX and PTX-SM-TAR NPs for 24 h, the cell nuclei were stained with DAPI (blue), and microtubules were stained with the Tubulin-Tracker Green (green). The scale bar is 30 μm. (**b**) Anti-proliferation effects of TAR peptide, PTX, PTX plus TAR peptide, and PTX-SM-TAR NPs at different concentrations against 4T1-mCherry-Luc cells (mean ± SEM, *n* = 3, * represents *p* < 0.05. (**c**,**d**) Statistical migration rates and images of 4T1-mCherry-Luc cells treated with different drug concentrations. The scale bar is 200 μm (mean ± SEM, *n* = 3, ** represents *p* < 0.01).

**Figure 9 ijms-24-04646-f009:**
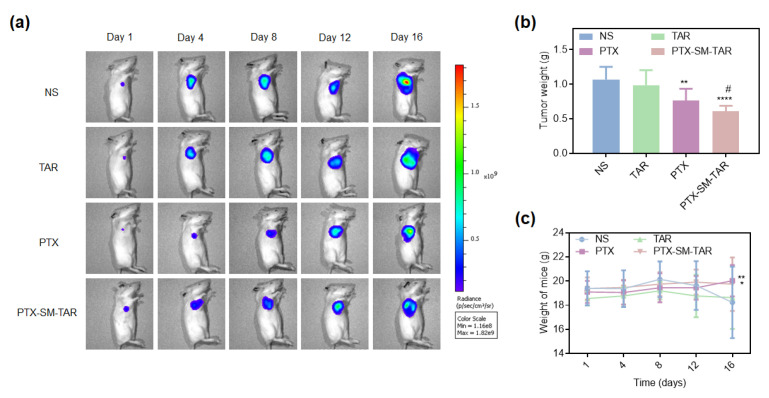
In vivo antitumor efficacy of PTX-SM-TAR NPs on 4T1-mCherry-Luc xenograft mice model (*n* = 7). (**a**) In vivo bioluminescence images of 4T1-mCherry-Luc tumor-bearing mice on 1, 4, 8, 12, and 16 d. (**b**) Weight of the excised tumors (mean ± SEM, *n* = 7, ** and **** represent *p* < 0.01 and *p* < 0.0001 versus the NS group, # represents *p* < 0.01 versus the PTX group). (**c**) Changes in mice’s body weight during the treatment process. * and ** represent *p* < 0.05 and *p* < 0.01.

**Figure 10 ijms-24-04646-f010:**
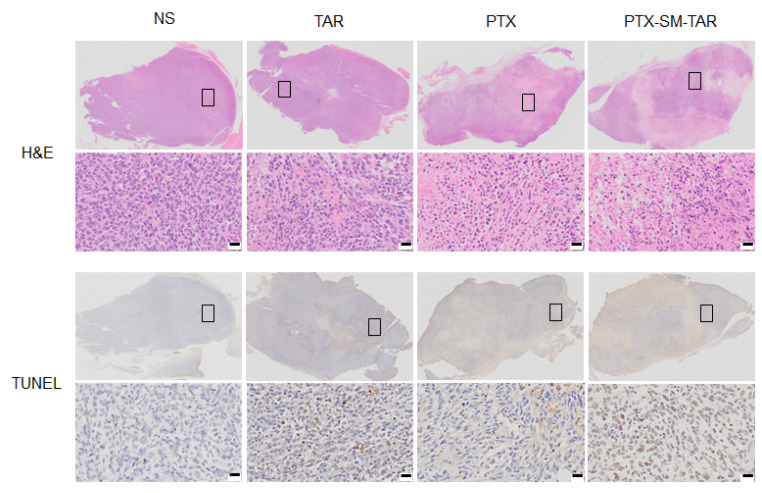
H&E and TUNEL staining images of tumor tissues at the end of treatment. The scale bar is 50 μm. The black box part in the upper image was enlarged by 400 times to obtain the corresponding local image below.

## Data Availability

Not applicable.

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
