# Peer review of "Transcytosable Peptide-Paclitaxel Prodrug Nanoparticle for Targeted Treatment of Triple-Negative Breast Cancer"

_ijms, 2023, doi:10.3390/ijms24054646_

Round 1

Reviewer 1 Report

1.It is suggested to add new research studies about the toxicological concerns for normal cells.
2.what is the suggestion of this study for future works?
3.Please discuss about the genetic factors against this nanosystem.
4.It will be better to add the role of mitochondria, NFkB-mediated apoptosis and gap junction proteins.

5.More references for the discussion part of manuscript and bold your study novelty should be added: e.g.,
-DOI: 10.3390/ph16010009
-DOI:10.1016/j.biopha.2018.01.117
-DOI:10.1007/s12032-022-01884-9

Reviewer 2 Report

1. In PTX-SM-TAR, what is SM?

2. Modify the order of subtitles of the results section. The "targeting and drug delivery properties of TAR peptide" should be transferred to the last section of the results.

3. In line 412, write the full name of ECGs.

4. The number of mice and their number in each group is not written

5. There are some relevant references that are recommended to be cited in the manuscript, such as

DOI: 10.1039/D1MA00961C

DOI: 10.3390/polym14040658

DOI: 10.1016/j.jconrel.2022.01.010

PMID: 27114800

DOI: 10.17305/bjbms.2016.674.

6. The manuscript should be updated with new relevant references (2022-2023).

Reviewer 3 Report

This work combined two targeting peptide to enhance the delivery of PTX. Overall, the authors organized the results nicely. But the novelty is not so clear. Vascular barrier is big problem of targeting tumor. EPR effect is always heterogeneous. The major advantage of this system is that it can improve the transcytosis beyond EPR effect. I strongly suggest the authors should emphasize this point (https://doi.org/10.1021/jacs.0c09029). At least, the evidence of transcytosis should be provided in vitro. 

Round 2

Reviewer 3 Report

The authors did some experiments to prove the transcytosis in vitro. Although no in vivo results were shown, I think it can be published. But please clearly discuss the advantage of transcytosis in introduction. I suggested the authors use 'transcytosable'. It is more appealing. Also the title can be modified, such as 'Transcytosable peptide-paclitaxel prodrug nanoparticle for targeted treatment of triple-negative breast cancer'. This reference may be useful for this work (https://doi.org/10.1021/jacs.0c09029).

Author Response

Response to Reviewer 3 Comments

Dear Reviewer:

Thank you so much for your professional and instructive suggestions for our manuscript entitled “A novel tumor targeting and penetrating peptide-drug conjugate-based prodrug nanoparticle for paclitaxel delivery in triple-negative breast cancer” (ID: ijms-2182495). Your comments are really valuable and helpful for improving our paper. We have studied these comments carefully and have made corresponding revisions. We hope these revisions could meet your requirements. These revisions are marked up using the “Track Changes” function in the manuscript. The point-by-point responses to the comments are listed below.

Point 1. The authors did some experiments to prove the transcytosis in vitro. Although no in vivo results were shown, I think it can be published. But please clearly discuss the advantage of transcytosis in introduction. I suggested the authors use 'transcytosable'. It is more appealing. Also the title can be modified, such as 'Transcytosable peptide-paclitaxel prodrug nanoparticle for targeted treatment of triple-negative breast cancer'. This reference may be useful for this work (https://doi.org/10.1021/jacs.0c09029).

Response 1: Thank you very much for your valuable suggestion. “Transcytosable” is indeed an attractive characteristic for this peptide-paclitaxel prodrug nanoparticle. According to your suggestion, the title has been changed to “Transcytosable peptide-paclitaxel prodrug nanoparticle for targeted treatment of triple-negative breast cancer”, which is more appealing than the original one. We also agree that it is necessary to clearly discuss the advantage of transcytosis in the introduction. We have added a paragraph to the introduction section for description, which is shown on page 3, lines 112-130. It is as follows.

In addition, the conjugate PTX-SM-TAR synthesized by hydrophilic TAR peptide and hydrophobic PTX is amphiphilicity and has the characteristics of self-assembly in an aqueous solution to form a core-shell nano-micelle. In antitumor therapy, nanomedicines mainly rely on the enhanced permeability and retention effect (EPR) to target the tumor tissue passively. However, the EPR effect is hindered by increased tumor interstitial fluid pressure, high viscous extracellular matrix, and tumor heterogeneity, which leads to the limitation of drug extravasation in blood vessels and insufficient penetration into the tumor entity to maximize the efficacy of nanomedicines[50]. Transcytosis is a broad transcellular transport process in which substances are transferred from one side of a cell to the other. Unlike passive diffusion, transcytosis is an active transport process that can deliver drugs through multiple pathways[51]. In recent years, transcytosable nanomedicine has become a new interest in promoting the penetration of nanomedicines in tumors[52,53]. The transcytosis design of these nanomedicines could depend on different forms, including receptor-mediated transcytosis, adsorption-mediated transcytosis and fluid-mediated transcytosis[51]. Based on the receptor-binding and positive charge properties of TAR peptides, we hypothesized that in addition to the EPR effect, the PTX-SM-TAR nanosystem can also be rapidly internalized by endothelial cells or tumor cells through receptor and adsorption-mediated transcytosis, improving extravasation efficiency in tumor vessels and enhancing permeability in tumor tissues.

Furthermore, we have also emphasized “transcytosis” in the abstract, keywords and conclusion sections, which are shown on page 1, lines 28-37, and page 18, lines 675-683. And the reference (https://doi.org/10.1021/jacs.0c09029) is very helpful to our manuscript, we have cited it in reference 51.

Round 3

Reviewer 3 Report

The authors addressed the concerns well. The story becomes more appealing.